# Is ß-d-glucan Relevant for the Diagnosis and Follow-Up of Intensive Care Patients with Yeast-Complicated Intra-Abdominal Infection?

**DOI:** 10.3390/jof8050487

**Published:** 2022-05-06

**Authors:** Hervé Dupont, Stéphanie Malaquin, Léonie Villeret, Pierre-Yves Macq, Nacim Ammenouche, François Tinturier, Momar Diouf, Matthieu Rumbach, Taieb Chouaki

**Affiliations:** 1Surgical ICU, Department of Anesthesiology and Critical Care Medicine, University Hospital of Amiens Picardy, 80054 Amiens, France; malaquin.stephanie@chu-amiens.fr (S.M.); villeret.leonie@chu-amiens.fr (L.V.); macq.pierre-yves@chu-amiens.fr (P.-Y.M.); ammenouche.nacim@chu-amiens.fr (N.A.); rumbach.matthieu@chu-amiens.fr (M.R.); 2SSPC Research Unit, University Jules Verne of Picardy, 80054 Amiens, France; 3Biostatistical Unit, Direction de la Recherche Clinique, University Hospital of Amiens Picardy, 80054 Amiens, France; diouf.momar@chu-amiens.fr; 4Mycology Laboratory, University Hospital of Amiens Picardy, 80054 Amiens, France; chouaki.taieb@chu-amiens.fr

**Keywords:** ß-d-glucan, intra-abdominal infection, candidiasis, intensive care unit

## Abstract

The usefulness of (1,3)-ß-d-glucan (BDG) detection for the diagnosis of intra-abdominal candidiasis and treatment monitoring is unknown. A prospective, single-center study of consecutive patients admitted to an ICU with complicated intra-abdominal infection (IAI) over a 2-year period was conducted. BDG was measured in the peritoneal fluid and serum between day 1 (D1) and D10. Patients with a positive peritoneal fluid yeast culture (YP) were compared to those with a negative yeast culture (YN). The evolution of serum BDG was compared in the two groups. Seventy patients were included (sixty-five analyzed): YP group (n = 19) and YN group (n = 46). Median peritoneal BDG concentration during surgery was 2890 pg.mL^−1^ [IQR: 942–12,326] in the YP group vs. 1202 pg.mL^−1^ [IQR: 317–4223] in the YN group (*p* = 0.13). Initial serum BDG concentration was 130 pg.mL^−1^ [IQR: 55–259] in the YP group vs. 88 pg.mL^−1^ [IQR: 44–296] in the YN group (*p* = 0.78). No difference in evolution of serum BDG concentrations was observed between the groups (*p* = 0.18). In conclusion, neither peritoneal BDG nor serum BDG appear to be good discriminating markers for the diagnosis of yeast IAI. In addition, monitoring the evolution of serum BDG in yeast IAI did not appear to be of any diagnostic value.

## 1. Introduction

Intra-abdominal infections (IAIs) are the second most common type of severe infection in the intensive care unit (ICU), with a high mortality rate reaching 30% [1]. In the vast majority of cases, IAIs are polymicrobial in origin. Yeasts are responsible for 13–30% of IAIs according to the literature [2,3,4,5,6]. Although yeast pathogenicity in IAI has been challenged, there is a consensus in favor of treatment, especially in the most severe cases, despite the lack of randomized clinical trials [7,8]. Isolation of yeasts has been associated with mortality in post-operative IAI [9] and, more recently, in community-acquired IAI [10]. However, at the bedside it is difficult to know which patient could benefit from empiric antifungal treatment. It would be possible to treat all ICU patients empirically with an IAI, but this is not reasonable for reasons of (i) patient security, (ii) economic aspects and (iii) antimicrobial stewardship (including antifungal).

Several approaches have been proposed to identify the patients who would benefit from antifungal treatment, including direct examination of peritoneal fluid for yeasts [11] and a predictive score suited to IAI [10,12]. (1-3)-ß-d-glucan (BDG), a component of the yeast cell wall, has been proposed for the diagnosis of candidemia, at a threshold of 80 pg.mL^−1^ for Fungitell^®^ assay [13,14,15,16]. More recently, the evolution of serum BDG levels was associated with the success or failure of antifungal treatment for candidemia [17]. However, data concerning IAI are scarce and difficult to interpret [18,19]. Furthermore, the use of BDG for the diagnosis of invasive candidiasis has a weak recommendation and a moderate degree of evidence [20].

The aim of this study was to evaluate serum BDG kinetics in a group of patients treated for yeast IAIs compared to a group of patients with yeast-negative IAI. Secondary evaluation criteria included an assessment of the diagnostic performance of initial serum and peritoneal BDG in yeast IAI.

## 2. Patients and Methods

### 2.1. Study Design

This was a prospective, observational, cohort study of consecutive patients hospitalized in the ICU with complicated IAI over a 2-year period. Patients with a yeast-positive peritoneal fluid culture (YP) were compared to those with a yeast-negative culture (YN).

### 2.2. Study Population

All consecutive adult patients hospitalized in the ICU after surgery with a diagnosis of complicated IAI were screened for eligibility. The definition of complicated IAI was made according to previous reports [21,22]. Briefly, complicated intra-abdominal infections extend beyond the source organ into the peritoneal space. They cause peritoneal inflammation and are associated with local or diffuse peritonitis. They can be community-acquired (e.g., appendicitis), nosocomial or post-operative. The diagnosis of complicated infection is made with an association of clinical signs (abdominal pain, fever or hypothermia, nausea or vomiting, etc.), biological signs (elevated or low white blood cell count), a CT scan showing localized or diffuse infection, and surgery with purulent peritoneal fluid coming from the source organ. They can be associated with organ failure.

Non-inclusion criteria included: known allergy to echinocandins; life-expectancy ≤48 h; withdrawal of care order; and red blood cell transfusion in the previous 3 months. Both groups of patients received antimicrobial treatment according to the recent consensus statement [7]. Antifungal treatment with an echinocandin (caspofungin) was given systematically to patients in the YP group.

### 2.3. Data Collection

The following data were recorded: medical history (Charlson score [23]); type of IAI (localized or generalized, community-acquired or nosocomial, location of the perforation, etiology). SAPS2 [24] and SOFA [25] scores were calculated at inclusion and the SOFA score was calculated daily until day 10 (D10). Daily routine laboratory results were noted until D10. Duration of mechanical ventilation, duration of catecholamine use, ICU length of stay, hospital length of stay, and outcome at D28 (dead or alive) were also recorded. Failure rate was calculated at the end of treatment (D10). This was defined as: any reoperation; death; no improvement of SOFA scores by D5; or modification of antimicrobial or antifungal treatment according to microbiological or mycological examinations.

### 2.4. (1,3)-ß-d-glucan Assays

A peritoneal sample and blood sample were taken during surgery, and blood samples were taken daily thereafter in a 5 mL dry tube until D10. The tubes were centrifuged at 8000 rpm for 10 min at 5 °C and the serum was aliquoted into Eppendorf^®^ tubes and stored in a freezer at −80 °C until BDG determination. Consumables used were cellulose-free and tested before the study. Clinicians were aware of the results because they were performed all together at the end of the study.

All (1-3)-BDG assays with Fungitell^®^ Trial (Associates of Cape Cod, Inc., East Falmouth, MA, USA) were performed at the same time at the end of the study. It was validated only for the serum sample. Proposed thresholds of the manufacturer are the following: <60 pg.mL^−1^, low probability of yeast infection; >80 pg.mL^−1^, high probability of infection.

### 2.5. Yeast Peritoneal Culture

The peritoneal fluid was collected just at the beginning of surgery before any intervention. It was sent directly to a mycological lab for culture. No other fluid was collected in drains. It was then seeded on 3 specific yeast media: (i) CHROMagar (Becton Dickinson Paris, France) at 35 °C, (ii) Sabouraud with chloramphenicol and gentamicin at 30 °C (Biomerieux, Lyon, France) and (iii) liquid Sabourand with chloramphenicol and gentamicin at 35 °C (Biomerieux, Lyon, France). The identification of fungi was performed by a MALDI-TOF Mass Spectrometry System (Bruker Daltonics^®^, Billerica, MA, USA) and in vitro antifungal susceptibility testing was determined by the Sensititre YeastOne^®^ (Thermo Fisher, Waltham, MA, USA).

### 2.6. Endpoints

The primary endpoint was the comparison of serum BDG kinetics between the YP and YN group. Secondary endpoints were the performance of initial serum and peritoneal BDG to diagnose yeast IAI.

### 2.7. Statistical Analysis

The sample size was calculated to obtain a Cohen effect size of 0.74 for the slopes of BDG between the groups [26]. Thirty patients per group were needed to demonstrate this effect, considered as medium to high according to Cohen’s classification.

The results are presented as mean ± standard deviation (SD) for variables with a normal distribution, and as median and interquartile range [IQR] for variables with a non-normal distribution. Qualitative variables are expressed as percentages with their 95% confidence intervals [CI]. Analysis of variance for repeated measures was used for the primary endpoint. Post-hoc analyses were performed using the Student’s *t*-test with Bonferroni correction. For the secondary endpoints, initial serum and peritoneal BDG levels were compared using the Wilcoxon test. An ROC curve analysis was carried out to evaluate the performance of serum and peritoneal BDG measurements for the diagnosis of yeast IAI. Demographic characteristics, laboratory test results, and outcomes were compared using the Student’s *t*-test or Wilcoxon tests depending on the distribution of the variables, and the chi^2^ test with Yates’ correction if necessary or Fischer’s exact test for qualitative variables. *p* < 0.05 was considered statistically significant.

## 3. Results

### 3.1. Study Population

Seventy patients were screened and five were excluded from the analysis (no peritoneal fluid examination (n = 3), refusal to participate (n = 2)). Sixty-five patients were therefore included in the final analysis (YP group (n = 19), YN group (n = 46)). Mean age was 65.6 ± 12.4 years and 38 (58.5%) were male. Two-thirds of the patients had nosocomial IAI and 73% had generalized infection. Overall mortality at hospital discharge was 29.3% (19/65). The demographic characteristics, type of infection, severity scores, clinical parameters, and main laboratory test results for the YP and YN groups are shown in Table 1. No significant difference was observed between the groups. Twenty-one strains of yeasts were isolated from the peritoneal fluid: 14 *Candida albicans*, 4 *C. glabrata*, 2 *C. tropicalis*, and 1 *C. krusei*. All these strains were susceptible to caspofungin. Only one patient in the YP group had a positive blood culture on admission to *C. albicans*.

### 3.2. Primary and Secondary Endpoints

No difference in the slope for BDG was observed between the YP and YN groups (*p* = 0.18) (Figure 1). Concerning the secondary endpoints, there was no significant difference in initial values for peritoneal BDG (1202.5 pg.mL^−1^ [IQR: 317.5–4223.5] YN group vs. 2890.5 pg.mL^−1^ [IQR: 942.5–12,326.5] YP group; *p* = 0.135) and serum BDG (88.0 pg.mL^−1^ [IQR: 44.5–296.0] YN group vs. 130.0 pg.mL^−1^ [IQR: 54.7–259.7] YP group; *p* = 0.785) between the two groups (Figure 2). A total of 2 consecutive serum BDG > 80 pg.mL^−1^ were observed in 15 patients in the YP group (78.9%) and 27 patients in the YN group (57.7%), *p* = 0.13. A single serum BDG > 200 pg.mL^−1^ was observed in 14 patients in the YP group (73.7%) and 28 in the YN group (60.9%), *p* = 0.33.

The ROC curve used to analyze the performance of these two tests for the diagnosis of YP IAI is shown in Figure 3. The area under the curve (AUC) was 0.621 [95% CI: 0.491–0.739] for peritoneal BDG (*p* = 0.11) and 0.52 [95% CI: 0.394–0.649] for serum BDG (*p* = 0.77).

The failure rate was 67.7%. The values of initial peritoneal BDG (2049.5 pg.mL^−1^ [IQR: 514.0–12,805.0] in the failure group vs. 840 pg.mL^−1^ [IQR: 203.0–4036.5] in the success group; *p* = 0.124) and initial serum BDG (154 pg.mL^−1^ [IQR: 54.5–310.5] in the failure group vs. 76 pg.mL^−1^ [IQR: 31.3–159.5] in the success group; *p* = 0.06) were not significantly different between the groups. The main outcomes are shown in Table 2. Briefly, mortality was not different between groups. However, the duration of mechanical ventilation and catecholamine use were shorter in the YN group than in the YP group. The success rate at the end of treatment was not significantly higher in the YN group (*p* = 0.08). There was no difference in initial serum and peritoneal BDG for in-hospital mortality: initial serum BDG was 129 pg.mL^−1^ [43–270] in the alive group and 93 pg.mL^−1^ [52–458] in the death group, *p* = 0.67; initial peritoneal BDG was 1228 pg.mL^−1^ [251–4953] in the alive group and 2207 pg.mL^−1^ [788–13,250] in the death group, *p* = 0.67.

## 4. Discussion

The main result of this study is the lack of any significant difference in the evolution of BDG between patients with YP IAI and those with YN IAI. Furthermore, neither initial peritoneal nor serum BDG were discriminating factors for the diagnosis of yeast IAI.

Available data about the usefulness of BDG for the diagnosis and monitoring of treatment in yeast-complicated IAI in ICU patients are scarce. The first published study evaluated the usefulness of serum BDG measurements to diagnose blood-culture-negative intra-abdominal candidiasis [19]. The authors found that two consecutive serum levels ≥80 pg.mL^−1^ had a sensitivity of 95% and specificity of 78% for the diagnosis of yeast IAI [19]. Furthermore, serum BDG was found to decrease in IAI responding to treatment, while it increased in patients with no response to treatment. The first concern was the great variability in serum BDG values whatever the group (uncolonized with *Candida*, colonized with *Candida* without antifungal treatment, colonized with *Candida* receiving preemptive antifungals, or documented yeast IAI). All groups had some patients with serum BDG ≤80 pg.mL^−1^ and some with serum BDG ≥80 pg.mL^−1^. It was impossible to know which group a patient was in using only BDG level. In our study, we did not find any difference in serum BDG between YP and YN patients, with serum BDG having a poor diagnostic performance from ROC curves. The slopes for the evolution of serum BDG levels from inclusion to D10 were not different in our study despite systematic antifungal treatment in the YP group. The second study published was a prospective randomized trial evaluating an antifungal drug for the preemptive treatment of patients with gastrointestinal surgery at risk of yeast IAI [27]. The rate of yeast IAI was low (10%), independent of antifungal treatment. However, the slope for serum BDG was positive in patients who will be yeast-infected when compared to a negative slope in those who will not be [27]. The third study was a small retrospective cohort of 33 patients with complicated IAI, of whom 7 were found to be YP [18]. These authors found a significant difference in peritoneal BDG between YP and YN patients (1461 pg.mL^−1^ in the YP group vs. 224 pg.mL^−1^ in the YN group). The values for BGD found in our study were much higher in both groups, without any significant difference. In the same way, the variability in peritoneal BDG in this third study was very high, as in our study. The diagnostic performance of BDG measurement could not be evaluated in such a retrospective study with few patients.

One of the main questions about our results is why we observed high peritoneal and serum BDG concentrations, even when yeast cultures were negative. Unlike candidemia, IAIs are often polymicrobial and may involve both bacteria and yeasts [3]. IAI is often due to a perforation of the gut, leading to fecal contamination of the peritoneum. The content of the gut could help us to understand our results. The healthy human mycobiome contains many fungal species, subject to environmental factors [28]. These have a close interaction with their host and dysbiosis of fungal communities has been associated with some diseases [29,30]. *Candida* species are frequently encountered [31]. However, many other fungal species have also been detected in the gut mycobiome: *Saccharomyces*, *Penicillium*, *Aspergillus*, *Cryptococcus*, *Malassezia*, *Cladosporium* spp. [28]. All of these species may colonize the gut at the same time and be a part of the inoculum in complicated IAI. Moreover, BDG is one of the most stable and common components of the cell wall of yeasts, molds, and dimorphic fungi [32,33]. This could explain why we observed such a high concentration in peritoneal fluid, and why it was higher in the YP group. For this reason, and unlike in candidemia, BDG is not relevant for the diagnosis of yeast IAI in polymycobial infections [13,14,34,35]. As the peritoneum is a semi-permeable membrane, the serum BDG is probably a reflection of peritoneum BDG with high concentrations in both groups.

No difference in mortality was observed between groups in the study. It was not designed to find some mortality differences. However, there was a tendency of higher serum BDG in the failure rate group and no difference was observed between initial serum and peritoneal BDG and mortality. It is difficult to conclude anything on secondary endpoints.

This study was a small, monocentric study. However, the calculated sample size was reached. All patients were consecutive patients under the same surgical team. The risk of misclassification between groups due to the sensitivity to peritoneal culture should be discussed. However, all peritoneal samples were sent to a mycological lab and were seeded on specific media to avoid false negative results. Furthermore, the sensitivity of peritoneal culture to yeast is clearly higher than that of blood culture [36]. A recent study did compare PCR approach to culture-based methods on the peritoneal fluid to detect yeasts’ pathogens. The overall agreement between the PCR assay and the culture method was good (*κ* = 0.79), and their sensitivities for the diagnosis of intraperitoneal candidiasis were 93.5% and 74.1%, respectively [36]. The measurement of BDG could be another concern. However, all peritoneal and serum specimens were centrifuged and frozen at −80 °C, and all analyses were performed at the same time at the end of the study with the same plates and reagents to limit variability in the analysis. The too-short timing of surveillance could be another bias. However, it is strictly the same as the duration of treatment we used. Another concern is about the treatment of yeast in complicated IAI. Currently, there are no prospective randomized trials on this topic, only retrospective studies with conflicting results. Clinicians should be aware that BDG is not a good marker for yeast IAI for the diagnosis and not follow the treatment with this marker.

## 5. Conclusions

In conclusion, neither peritoneal BDG nor serum BDG appear to be good discriminating markers for the diagnosis of yeast IAIs. Likewise, monitoring the evolution of serum BDG does not appear to be useful in yeast IAI. Future research should focus on new specific biomarkers for *Candida* infection rather than on panfungal biomarkers.

## Figures and Tables

**Figure 1 jof-08-00487-f001:**
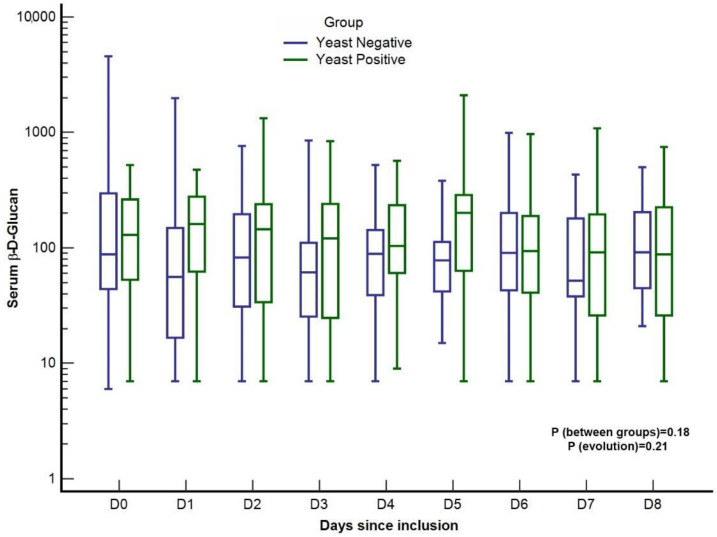
Time course of serum (1,3)-ß-d-glucan concentrations according to yeast positivity of peritoneal fluid in patients with complicated intra-abdominal infections. Values are in pg.mL^−1^.

**Figure 2 jof-08-00487-f002:**
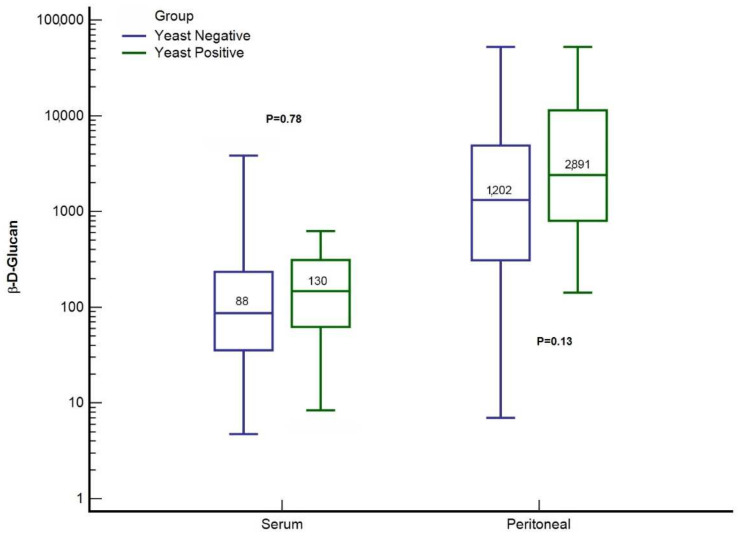
Initial serum and peritoneal (1,3)-ß-d-glucan concentrations in patients with complicated intra-abdominal infections. Values are in pg.mL^−1^.

**Figure 3 jof-08-00487-f003:**
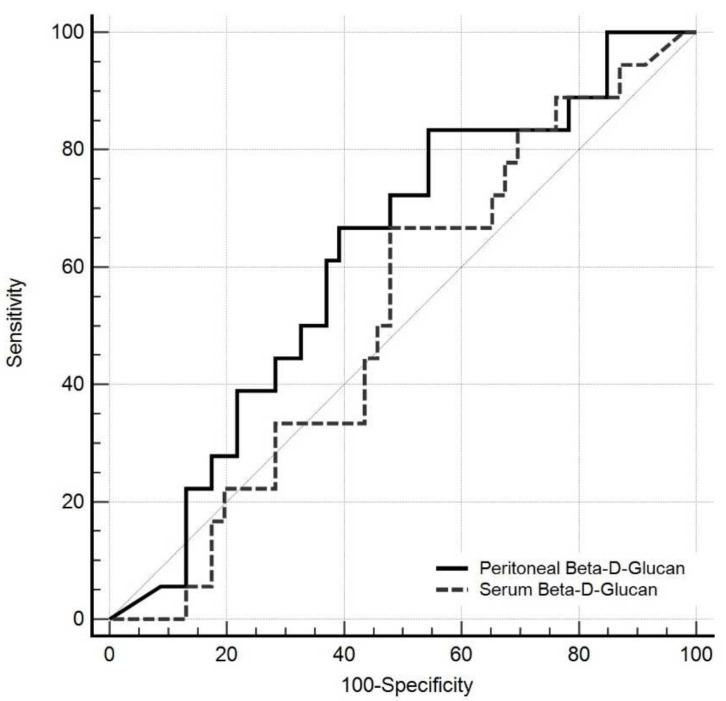
ROC curves of initial serum and peritoneal (1,3)-ß-d-glucan concentrations to diagnose intra-abdominal candidiasis.

**Table 1 jof-08-00487-t001:** Main characteristics of the study population according to peritoneal fluid yeast culture result.

	Yeast Negative Peritoneal Culture (n = 46)	Yeast Positive Peritoneal Culture (n = 19)	*p* Value
Age (years)	69 [59–73]	62 [59–71]	0.42
Charlson score	5.0 [3.5–7.0]	5.0 [4.0–6.5]	0.83
Sex, male	25 (53.2%)	14 (73.7%)	0.21
SAPS2 score	65 [51–74]	66 [64–78]	0.48
SOFA score	14.0 [11.5–16.0]	15.0 [11.5–16.0]	0.29
Community IAI	16 (34.0%)	5 (26.3%)	0.65
Generalized IAI	33 (70.2%)	15 (78.9%)	0.68
Upper gastro-intestinal tract location	14 (29.8%)	10 (52.6%)	0.35
MAP (mmHg)	82 [70–90]	77 [74–88]	0.42
HR (bpm)	100 [85–123]	103 [90–114]	0.97
Mechanical ventilation	43 (91.5%)	17 (89.5%)	0.99
Procalcitonin (µg.L^−1^)	3.6 [0.8–21.0]	4.9 [0.9–25.3]	0.89
CRP (mg.L^−1^)	221 [153–299]	150 [52–322]	0.23

Values shown are median [interquartile range] or n (%). SAPS2: simplified acute physiologic score; SOFA: sepsis-related organ failure assessment; IAI: intra-abdominal infection; MAP: mean arterial pressure; HR: heart rate; CRP: C-reactive protein.

**Table 2 jof-08-00487-t002:** Outcome of the patients according to peritoneal fluid yeast culture result.

	Yeast Negative Peritoneal Culture (n = 46)	Yeast Positive Peritoneal Culture (n = 19)	*p* Value
Duration of catecholamine use (days)	3 [2–5]	5 [3–9]	0.005
Duration of MV (days)	3 [1–6]	6 [4–16.5]	0.001
Success rate at EOT	18 (39.1%)	3 (15.8%)	0.085
Mortality at day 28	10 (21.7%)	4 (21.5%)	0.95
Mortality at hospital discharge	14 (30.4%)	5 (26.3%)	0.74

Values shown are median [interquartile range] or n (%). MV: mechanical ventilation; EOT: end of treatment (day 10).

## Data Availability

All data analyzed during the current study are available by requesting the corresponding author.

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
