# Peer review of "Is ß-d-glucan Relevant for the Diagnosis and Follow-Up of Intensive Care Patients with Yeast-Complicated Intra-Abdominal Infection?"

_jof, 2022, doi:10.3390/jof8050487_

Round 1
Reviewer 1 Report
In this manuscript Dupont and colleagues present the results of an single center prospective study on BDG levels in peritoneal fluid and serum of patients with intraabdominal infections.
Its conclusion are in line with the results and are very important to readers who use BDG to guide antifungal therapy of intraabdominal infection.
The manuscript is easy to follow and well written. However, there are some points that I would suggest to be addressed prior to publication.
Minor point:
Abstract, line 22: "grou" should read as "group"
Introduction: Line 41/42: Please rephrase: I guess it would be possible to treat all patients but it is not reasonable for reasons of i) patients security and ii) economic aspects and iii) antimicrobial stewardsship (including antifungals). The latter is adressed by "ecological reasons". However, in my view, this is term correct but its sense might not be recognized by all readers.
Line 46: This breakpoints only applies to the fungitell assay. As there are other commercial tests availible (Wako), please indicate or discuss breakpoints of other platforms.
Please include briefly in the introduction: The fungitell test is certified for the detection of BDG in serum samples.
Methods:
Line 64: Which antibiotic therapy was mainly used? E.g. sulbactam is suspected to produce false positve BDG results (doi:10.3390/jof7010014)
Line 77-80: I suggest to include the statement, that the consumables used are cellulose-free, as this is the major source of contamination of BDG testing in the laboratory.
Line 81 to 87: The start of this paragraph can be deleted. The mechanism of action of the fungitell assay is not relevant to the study. A statement of the use according to the manufacturers instruction or any alteration to it would be satisfying.
Line 96 - 97: I am not an expert in statistics. However, please clarify: the sentence states that 30 individuals per group are needed for the requested effect size but the YP group includes 19 patients.
Figure 1:
X-Axis: Please geplace J(our)1-10 by d1-10
Y- Axis: beta or β - D - glucan, the log scale should read as in figure 2 (10/100/1000/...)
Discussion:
Line 184-198: Please include a brief discussion of the elevated serum levels of both groups, too.
Please briefly discuss the insignificant difference of Mortality (28 days and end of stay). I m view, the surgical course of disease but not the infection are relevant to these endpoints.
Major point:
Please include the definition of intraabdominal infection in the section "study population".
Author Response
Please see the file attachment

Reviewer 2 Report
A complete list of my comments is provided in the attached table. However, the main observation to the results of the study is as follows:
The results of the study are in contrast to the literature data recognizing the BDG test as having high negative predictive value. In fact, in the study even YN patients present high serum BDG values throughout the observation period. The presence of many fungal species in the gut microbiome, as commented by the authors, does not seem sufficient to explain the positive serum BDG results in YN patients. Therefore, a suggestion to add value to the results reported is to assess whether two consecutive values of serum BDG > 80 pg/ml or a single value of serum BDG > 200 pg/ml are significantly associated with IC in YP.

Author Response
Please see the file attachment

Reviewer 3 Report
In this draft, the authors described the usefulness of serum or peritoneal fluid beta-D-glucan (BDG) in IAI patients. As the authors mentioned, BDG is a component of fungi, therefore, serum of peritoneal fluid BDG might be higher in fungal IAI patients, however, the authors showed that serum and peritoneal fluid BDG was not statistically different between Yeast-positive group and Yeast-negative group. According to these data, serum or peritoneal BDG was not useful marker of diagnosis or follow-up of IAI. However, there are lacking of several important data in this draft.
The most concerning point is the definition of IAI related with fungi or not (Yeast-positive or Yeast-negative group). The authors classified these group according to the results of peritoneal fluid culture. However, peritoneal fluid was collected only during operation. The sensitivity of fungal culture is known to be relatively low, therefore, this only one sample collection could be linked to underdiagnosis of fungal IAI. The authors also mentioned that some samples in yeast-negative group showed higher BDG concentration than those in yeast-positive group, however, only one sample collection could not pick up all of the true “yeast-positive” IAI patients. This point is the big concern and could be related with the results that there were no significant differences between yeast-positive and yeast-negative group. Their conclusion might be misled by this "only one point sample collection".
The other concerning point is that there were few messages to readers from this study. The authors mentioned serum or peritoneal fluid BDG was not useful to diagnose of fungal IAI, however, the diagnosis of IAI depended on the culture results. If serum or peritoneal fluid BDG was high in some patients, clinicians did not change their treatment. However, some patients in whom BDG might have fungal IAI, therefore, antifungal treatment for these patients might be beneficial, even if the fungal culture results of peritoneal fluid was negative. The authors should discuss these points.
The other major points were as follows:
- The results of culture:The authors did not show the isolates of Yeast-positive group. The authors described that echinocandin was used as empirical therapy for IAI, however, several fungi were known to be resistant to echinocandin. If the authors would like to mention failure rate, the isolates of Yeast-positive group is quite important data. Therefore, the authors should provide these data, in addition drug susceptibility data.
- Culture methods:Along with point 2, the authors should provide the culture methods of peritoneal fluid. Used culture media / agar affects culture results. The authors mentioned that peritoneal fluid BDG is high in some patients of Yeast-negative group, however, some fungal isolates might be missed due to culture methods. To clarify these points, culture methods should be added in detail.
- Definition of IAI:The authors should clarify the definition of IAI. If possible, the authors should cite papers of the definition of IAI.
- The value of Figure 1 and Figure 2:In Figure 1, the authors showed the value and trends of serum BDG in Yeast-positive and Yeast-negative groups. However, the value of BDG was shown as 88.0 pg.ml-1 [IQR: 128 44.5‒296.0] YN group vs. 130.0 pg.ml-1 [IQR: 54.7‒259.7] YP group in the manuscript. What Log of the serum BDG level means in Figure 1? (The value of Log 4 means serum BDG exceed 10,000 pg.ml-1). The authors should clarify this point. In addition, the authors used Log [Serum of peritoneal BDG] in vertical scale, however, vertical scales is not log in Figure 2. If the authors would like to use Log scale, vertical scales should be shown as 1, 2, 3.
- The meaning of Table 2:The authors would like to describe the usefulness of serum or peritoneal fluid BDG in this paper, however, Table 2 did not provide any data related with this point. In addition, the authors did not mention the relationships between this outcome and serum/peritoneal BDG. The authors should discuss this data in detail.
The authors described the insufficient usefulness of BDG of serum or peritoneal fluid, but the data are very insufficient to discuss the usefulness of BDG. It is misleading to make this conclusion from the data presented, and it is considered necessary to review the data as a whole.
Author Response
Please see the file attachment

Reviewer 4 Report
- A brief summary The authors evaluated usefulness of (1,3)-ß-D-glucan (BDG) detection for diagnosis and treatment monitoring of intra-abdominal infections (IAI). Study on 65 consecutive patients with IAI admitted to ICU over 2-year period showed that neither peritoneal nor serum BDG appear to be good discriminating marker for the diagnosis of yeast IAI. Also, evolution of serum BDG did not show any diagnostic value.
-
General concept comments
Review: This manuscript describes clear and relevant authors' observations and is well presented. Study design is appropriate, tables and figures are easy to understand and interpret and the conclusion is consistent. However, reference list should be improved with newer references (24/32 references are more than five years old, i.e. published before 2017)
- Specific comment: line 46 - authors wrote that BDG, a component of the yeast cell wall, has been proposed for the diagnosis of candidemia, at the threshold of 80 pg/mL. This statement should be corrected because the threshold of 80 pg/mL is specific for only one kit (Fungitell). Although this kit is the most widely used test for BDG, there are also other kits for detection of BDG that are used with different thresholds. References number 13-16 cited here are also referring only to Fungitell kit and the authors should also add the references about other existing kits and their positive cut-off values should be mentioned as well; if there are studies with other kits used in patients with IAI, they should be discussed in the Discussion section; line 76 - the authors used Fungitell kit which is validated for use only on serum samples and in this study it was used on both serum samples and peritoneal fluid; this fact should be said in the Patients and Methods section; line 139 - results in Table 2 about outcome of the patients according to peritoneal yeast culture result should be described in the Result section (are there any significant differencies between two groups?); these results should also be commented in the Discussion section
Author Response
Reviewer N°4
We thank the reviewer for the comments about the manuscript.
Please find enclosed a point-to-point response to the comments.
- Old references. We agree with the reviewer comment. However, recent publications on this topic are quite scarce. That is one of the reasons that old references are used.
- Line 46: We agree with the reviewer comment and a specific information was written in the method section about the Fungitell kit. To the best of our knowledge, in the published papers for yeast IAI, no other kit was used. (p5 ln24-25 and p6 ln1-3)
- Line 76: according to the reviewer comment, this has been added in the materials and methods section.(p6 ln 1)
- The table was described in the results section and discussed. (p8 ln 18-25, p10 ln 24-25 and p11 ln 1-2)
Round 2
Reviewer 3 Report
The authors revised and resubmitted the manuscript according to the reviewers' comments. Some of the contents are considered to be improved, but there are some sections that need further improvement and additional information.
Major Points:
- The authors argued against the definition of YP and YN groups, in line with the peer review comments. Although the sensitivity of peritoneal fluid culture has not been reported so far, the authors' statement that "the sensitivity of peritoneal culture to yeast is clearly higher thah (should be collected as than) that of blood culture" must be supported by evidence or papers. If the authors would like to suggest that the definition of YP and YN groups is not problematic because of the sensitivity of peritoneal culture, the authors have to provide evidence. If this evidence does not exist, the sensitivity of peritoneal fluid remains questionable, and the authors should reconsider the implications of the lack of difference in BDG when compared separately to YP and YN groups.
- Although the authors stated that "Species identification and susceptibility to antifungal were performed.", information on this detail is needed (method of identification and drug susceptibility tests). Identification and drug susceptibility test of fungi is complicated, therefore, some methods might incorrectly determine susceptible/resistance. Additional information on this point is essential.
Author Response
We thank the reviewer for the additional comments.
- In fact, the culture method remains the gold standard for the diagnosis of intraabdominal candidiasis. We found a recent study (Corralese et al. ,Medical Mycology, 2015, 53, 199–204 doi: 10.1093/mmy/myu075) comparing PCR approach to culture based method on the peritoneal fluid to detect yeasts pathogens. The overall agreement between the PCR assay and the culture method was good (κ = 790), and their sensitivities for the diagnosis of intraperitoneal candidiasis were 93.5% and 74.1%, respectively. This point has been added with the new reference in the discussion on the limit of the study. (p11, ln 2-6)
- The identification of fungi was performed by MALDI-TOF Mass Spectrometry System (Bruker Daltonics®, Billerica, USA) and In vitro antifungal susceptibility testing was determined by the Sensititre YeastOne®Thermo Fisher, Waltham, USA). This information is now added in the manuscript in the material and methods section. (p6, ln 4-6)